# Sensitivity, Specificity, and Predictive Values of Tru-Cut^®^ Biopsy in Grading Primary Localized Myxoid Liposarcomas of the Extremities

**DOI:** 10.3390/cancers15051391

**Published:** 2023-02-22

**Authors:** Giuseppe Bianchi, Roberta Laranga, Paolo Spinnato, Federico Ostetto, Elisa Bubbico, Alberto Righi, Davide Maria Donati

**Affiliations:** 13rd Orthopaedic and Traumatologic Clinic Prevalently Oncologic, IRCCS Istituto Ortopedico Rizzoli, Via Pupilli 1, 40136 Bologna, Italy; 2Diagnostic and Interventional Radiology, IRCCS Istituto Ortopedico Rizzoli, Via Pupilli 1, 40136 Bologna, Italy; 3Anatomy and Pathological Histology, IRCCS Istituto Ortopedico Rizzoli, Via di Barbiano 1/10, 40136 Bologna, Italy; 4Department of Biomedical and Neuromotor Sciences (DIBINEM), Alma Mater Studiorum University of Bologna, 40123 Bologna, Italy

**Keywords:** soft tissue neoplasms, myxoid liposarcoma, pathology, biopsy, diagnosis, histological grade, diagnostic accuracy

## Abstract

**Simple Summary:**

Biopsy is an essential step in the diagnosis of myxoid liposarcoma (MLs) since the histological grade is a strong determinant of the appropriate treatment in the management of this pathology. The aim of our retrospective study was to investigate the diagnostic accuracy of Tru-cut^®^ biopsy (TCB) and the potential impact on a patient’s survival in the case of misdiagnosis. We established that in MLs, diagnosis with TCB might differ from that of surgical specimens, with a histological grade concordance rate of 64% (Kappa 0.30). Neoadjuvant treatments were associated with pathological downgrading with a lower effect of chemotherapy alone compared to neoadjuvant-combined treatments, although such discordance did not modify the prognosis. In the group of patients not treated in neoadjuvant settings, the sensitivity and specificity of TCB were 57% and 100%, respectively. TCB results are useful in leading the clinician in the diagnostic pathway thought; the diagnosis of MLs should be supported by other diagnostic techniques.

**Abstract:**

(1) Background: Histological diagnosis and tumor grading are major prognostic and predictive factors in soft tissue sarcomas (STS), as they dictate the treatment strategies with a direct impact on patient survival. This study aims to investigate the grading accuracy, sensitivity, and specificity of Tru-Cut^®^ biopsy (TCB) in primary localized myxoid liposarcomas (MLs) of the extremities and its impact on patient prognosis. (2) Methods: Patients with ML undergoing TCB and a subsequent tumor resection between 2007 and 2021 were evaluated. Concordance between the preoperative assessment and definitive histology was calculated with a weighted Cohen’s kappa coefficient. Sensitivity, specificity, and diagnostic accuracy were calculated. (3) Results: Of 144 biopsies, the histological grade concordance rate was 63% (Kappa 0.2819). Neoadjuvant chemotherapy and/or radiotherapy impacted concordance with a downgrading effect in high-grade tumors. Among forty patients not treated in neoadjuvant settings, the sensitivity of TCB was 57%, the specificity was 100%, and the overall predictive values of positive and negative TCB were 100% and 50%, respectively. Misdiagnosis did not impact overall survival. (4) Conclusions: TCB may underestimate ML grading due to tumor heterogeneity. Neoadjuvant ChT and/or radiotherapy are associated with pathological downgrading; however, discordance in diagnosis does not modify patient prognosis because systemic treatment decision-making also includes other variables.

## 1. Introduction

Myxoid liposarcoma (ML) is the second most frequent subtype of liposarcoma, a malignant mesenchymal tumor originating in adipose tissue, presenting in 15–20% of cases and accounting for about 5% of all soft tissue sarcomas in adults [1].

Epidemiologically, MLs are most frequent in the fourth and fifth decades of life, without differences between the two sexes, and they are also the most widespread subtype of liposarcoma in children and adolescents [2]. The deep tissues of the extremities are the sites where ML localizes with the highest incidence, particularly in the thigh. In rare cases, they may localize subcutaneously and retroperitoneally [3]. Patients affected by ML generally have a good prognosis, with an estimated 5-year overall survival rate of 78–96% in localized tumors [4,5,6,7]. Local recurrences occur in 12–25% of cases, while distant metastases afflict 30–60% of patients, presenting even years after diagnosis. The regions mainly involved in metastasis are the bones, lymph nodes, lungs, and abdomen [4,7].

Treatment of ML consists of a combination of surgery and radiotherapy (RT) associated with chemotherapy (ChT), according to clinical presentation. ML is highly sensitive to RT and partly sensitive to ChT [5,8].

Histologically, ML is characterized by uniform, oval to round cells, with a variable number of small lipoblasts, set in a myxoid stroma, with a capillary-sized vascular network, organized in a distinctive plexiform pattern previously called chicken wire or crow’s feet. The presence of FUS-DDIT3, or less commonly, EWSR1-DDIT3, is pathognomonic to this entity [9,10]. Different threshold values of hypercellular areas (also known as round cell differentiation when neoplastic cells assume round cell morphological features) ranging from 5–25% have been set by different studies [3,10,11,12,13,14]. The WHO’s classification of soft tissue and bone tumors recommends that any amount of hypercellularity should be reported in the pathological report, and if it exceeds 5%, the tumor should be considered high-grade [15]. The presence of >5% hypercellularity is associated with a significantly poorer prognosis, identifying patients at high risk, which is a determining factor in treatment planning [5,8,16]. Pathological evaluation with molecular confirmation of specific fusion transcripts is necessary to define the histologic subtype and grade; it should be obtained before definitive treatment in the case of a suspected ML. The routine procedure for pathological diagnosis involves multiple preoperative core needle biopsies, generally using 14–16 gauge needles. [11,17]. Tru-Cut^®^ biopsy (TCB) is one of the most commonly used core needle procedures. It is a minimally invasive technique that can be easily performed under ultrasound guidance and with local anesthesia. The advantages of TCB include cost effectiveness, avoidance of diagnostic delays, low complication rates, and minimal invasiveness [18]. Enough viable tissue representative of the lesion and available for histopathological and immunohistochemical evaluation is required. However, a core biopsy may not always accurately reflect the histological features of the tumor. It provides a relatively small sample of tumor tissue, and, taking into account intrinsic tumor heterogeneity, obtaining a representative sample can be difficult [19,20,21]. This could result in a diagnostic inaccuracy in the form of underestimating the grade in patients with ML (downgrading error) and could result in a different treatment decision-making process, including the surgical approach and neoadjuvant ChT/RT [18,22,23].

To our knowledge, no studies have reported the correlation between the discordance associated with tumor grade in ML and the impact on treatment planning. Therefore, this study aims to report our institutional experience and investigate the diagnostic accuracy, sensitivity, and specificity of the histologic TCB procedure and the potential impact of misdiagnosis on patient survival.

## 2. Methods

### 2.1. Patients and Methods

The study was approved by the Institutional Review Board of the Rizzoli Orthopaedic Institute of Bologna (protocol code: CE AVEC 58/2022/Oss/IOR; date of approval: 14 February 2022).

We retrospectively reviewed the histological and clinical records of 150 patients affected by myxoid liposarcoma treated between 2007 and 2021 and selected 144 cases of primary, localized MLs of the extremities with molecular confirmation of the diagnosis. Exclusion criteria included patients with metastasis at the onset or with a non-histologically proven diagnosis. Hypercellularity of greater than 5% was used as the threshold for labeling the ML high grade [7,24]. The results of the grade determined in the core biopsy were compared to the complete resection final pathology reports. Follow-up data, including the status of patients at the last visit, were updated. Clinical data included patient demographics (age, gender), tumor properties (site, size, depth, and histology), diagnostic and therapeutic regimens (type of surgery, surgical margins, neoadjuvant and adjuvant therapy), and clinical outcome (status, local recurrence, and distant metastasis after treatment. Tumor size was assessed with a preoperative MRI. The histological diagnosis was confirmed by TCB. Outcome variables of determining malignancy, determining exact diagnosis, and treatment for core biopsy were measured against the final clinical diagnosis performed by analysis of the completely resected specimen in combination with the final clinical impression.

### 2.2. Biopsy Procedure

Biopsy procedures were performed after a careful evaluation of MRI studies by orthopedic surgeons and radiologists with the aim of choosing the best approach. According to a standardized protocol, a percutaneous core needle biopsy was performed on the tumor by the orthopedic and radiologist oncology team. The core biopsies were performed using a 14-gauge Tru-Cut^®^ soft tissue biopsy needle (Cardinal Health, Dublin, OH, USA) through the insertion site under ultrasound guidance, taking multiple samples (3–5 passes) throughout the tumor circumferentially, being careful to obtain adequate tissue for evaluation but not breach the far wall of the tumor. Power/color Doppler evaluation was performed during pre- and intra-procedural ultrasounds to avoid necrotic areas within the biopsy sample.

### 2.3. Therapeutic Procedures

Patients were treated using a multimodality approach, including surgery, RT, and ChT. The choice of every surgical procedure was performed in an effort to obtain the broadest oncological margins. Surgical margins were defined according to the Musculoskeletal Tumor Society, based on Ennekin et al. classification [25]: negative margins (microscopically negative) are indicated as R0-wide/radical; in case of the involvement of margins, a distinction is drawn between complete macroscopic resection with microscopic involvement (R1-marginal) and incomplete macroscopic resection (R2-intralesional). RT was administered according to the most appropriate technique available: in the preoperative setting, with a total dose of 50 Gy in 1.8–2 Gy fractions, and in the postoperative setting, with doses up to 66 Gy, based on the presentation, age, and resection margins. ChT treatment consisted of the combination of doxorubicin and ifosfamide or epirubicin and ifosfamide or trabectedin in the neoadjuvant and adjuvant setting. The surgery took place 4 weeks after the termination of the last cycle of ChT or the last fraction of RT [17]. Follow-up procedures consisted of clinical examination and MRI with contrast enhancement of the primary tumor site and a chest CT (every 3 months for the first 2 years, every 4 months during the third year, every 6 months for the fourth and fifth years, and annually from years 6 through 10). Abdomen CT scan with contrast enhancement was performed every 6 months for the first 2 years, every 8 months during the third year, and annually during the rest of the follow-up until the tenth year.

### 2.4. Statistical Analysis

The description of quantitative variables was performed using median and range. The qualitative variables were presented by means of the description of proportions. A comparison between the presurgical biopsy and postoperative histological analysis was performed. The correlation between groups was calculated using Cohen’s Kappa coefficient. Complete agreement is considered as a Kappa score of 1. Kappa values close to or less than 0 show poor correlation. Sensitivity, specificity, positive predictive value, negative predictive value, and the overall accuracy of the clinical tests were calculated with the two-by-two table method. The final histology of the resected (surgical) specimen was used as the reference standard, and two-by-two contingency tables were reconstructed for TCB for the final histological diagnosis. A forward stepwise logistic regression analysis was used to determine the risk of error in predicting the various grades of ML. The concordance with the final diagnosis, as confirmed by postsurgical biopsy, was used as the dependent variable. The level of significance for each clinical test was set at 0.05. Statistical analysis was performed using STATA Software (version 17).

## 3. Results

### 3.1. Clinicopathological Features

Our data included 144 localized myxoid liposarcomas (ML). Table 1 summarizes the main clinicopathological features of the patients and of the tumors analyzed. In our study, we evaluated 91 men and 53 women. The mean age at diagnosis was 48 years (16–78). Regarding tumor localization, 100 (69%) ML were found in the thigh, 33 (21%) in the lower limb, 10 (7%) in the buttock, and 1 (1%) in the arm. Most of the tumors (136 or 95%) were deep-sited. The size, determined by preoperative MRI, was larger than 10 cm in 83 (58%) patients, between 5 and 10 cm in 55 (38%) patients, and smaller than 5 cm in 6 (4%) patients. At the preoperative biopsy, 87 (60%) tumors were classified as high grade (>5% round cells), while 57 (40%) were low grade. Most patients were treated with surgical excision (99% of cases), while only two patients were treated with amputation due to the size and location (lower limb and thigh) of the tumor. A final histopathological diagnosis on the resected tissue specimens confirmed that 59 tumors (41%) were high grade and 85 (59%) were low grade. In overall cases, 104 (72%) resection margins were R0, 35 (25%) were R1, and 5 (3%) were R2. In cases of amputation (2/144; 2%), surgical margins were radical.

Neoadjuvant RT was administered in 92 ML—59 high-grade and 33 low-grade (59 cases > 10 cm, 29 cases 5–10 cm, and 4 cases < 5 cm)—while postoperative RT was performed in 31 patients—20 high-grade and 11 low grade (18 cases > 10 cm, 13 cases 5–10 cm, and none < 5 cm). Forty-six (32%) ML patients were treated with neoadjuvant ChT, while 20 (14%) patients had postoperative ChT, and 2 (1%) patients had both pre- and postoperative treatment. Thirty-six and twelve patients were treated with the combination of ChT and RT in a neoadjuvant or adjuvant setting, respectively.

Follow-up data were available for all patients. The median follow-up was 69 months (range 2–158 months). At the end of the study, 4% had died of cancer (6/144), 2% had died of unknown causes (3/144), 85% were alive without disease (122/144), and 9% were alive with disease (13/144).

### 3.2. Determination of Histologic Accuracy: Cytohistologic Correlation of Grade Diagnosis

Following TCB, the 144 cases were grouped into two categories: Group A was high grade (*n* =  87), and Group B was low grade (*n* = 57). The histologic post-resection diagnosis of these cases was 59 high grades and 85 low grades. Specifically, there was a downgrading in 40 cases of Group A and an upgrading in 12 cases belonging to Group B. Details are reported in Table 2. Overall, the concordance rate between the two biopsies was 64% (Kappa 0.30).

### 3.3. Risk Factors Associated with Downgrading

Logistical data analysis was performed to examine potential factors contributing to downgrading in Group A (Table 3). Neoadjuvant therapy was associated with pathological downgrading. Specifically, patients treated with neoadjuvant RT had a higher probability of downgrading compared to patients treated with neoadjuvant ChT (OR 4.66; 95% CI 0.09–0.21 *p* < 0.052). The trend was confirmed for patients treated with a combination of chemotherapy and RT (OR 5.30 *p* = 0.026) in comparison to only ChT. No confounding effect associated with measuring association was found after adjusting for the other variables in multiple regression analysis.

### 3.4. Impact of Misdiagnosis on Prognosis

Of the 144 cases analyzed, the overall survival curves and estimation are reported in Figure 1 and Table 4.

Considering 87 patients resulted in a high-grade at-core needle biopsy (Group 1), 45 “real” low-grade tumors (Group 2), and 12 undergoing “upgrade” (Group 3), misdiagnosis had no significant impact on prognosis (Figure 2A,B, Table 5).

### 3.5. Concordance Rate in the Group of Patients Not Treated

We focused our study on the sub-cohort of patients treated without neoadjuvant therapies to explore grade rate concordance between the two biopsies without any confounding effect. A total of 40 ML patients were analyzed. Following TCB, upon histologic examination, the 40 cases were grouped into high-grade (*n* = 16) and low-grade (*n* = 24) groups. The histologic post-resection diagnoses of these cases confirmed 28 high grades and 12 low grades. Twelve cases did not show concordance with the final excisional biopsy and were, therefore, underestimated in the previous analysis (Table 6). The overall sensitivity of TCB was 57%. The overall specificity was 100%. The overall predictive value of a positive TCB was 100%, and the overall predictive value of a negative TCB was 50% (Table 6). No potential factors for grading errors were found.

## 4. Discussion

Tumor grading assessment is critical in defining the best therapeutic approach [26,27] in soft tissue sarcomas. The most common grading system, the FNCLCC (Fédération Nationale des Centres de Lutte Contre le Cancer) classification, defined by a combination of tumor differentiation, mitotic count, and necrosis, remains controversial in grading myxoid liposarcomas and other specific histology. The presence of hypercellularity or round cell differentiation is linked to a worse prognosis [7,16] in myxoid liposarcomas. However, different threshold values, ranging between 5% and 25%, have been set by independent studies [3,10,28,29]. According to the WHO’s classification of soft tissue and bone tumors, in the pathological report, any amount of hypercellularity should be reported; if it exceeds 5%, the tumor should be considered high-grade [11].

TCB under ultrasound guidance with multiple tissue samples (14–16 gauge needles) is widely accepted as the gold standard for tumor sampling and diagnosis [13,30]; however, biopsy specimens do not always represent the entire tumor heterogeneity [12,30]. Especially in myxoid liposarcomas, the presence of transitional areas between typical low-grade histology with modestly increased cellularity and high-grade round cell morphology showing cellular overlap, elevated nuclear grade, mitotic activity, and obscuring of the underlying vascular pattern can be confusing in evaluating the percentage of hypercellularity [13]. Core needle biopsy accuracy, in combination with ultrasound guidance, could be useful to obtain the most representative samples of pathological tissue avoiding “blind” sampling techniques or incisional biopsy [22,31,32,33,34,35]. The combination of these factors entails potential misdiagnosis of myxoid liposarcomas with a hypothetical impact on the patient’s clinical history.

In this study, we assessed a concordance rate of 64% between the biopsy and the final pathological report. In the low-grade TCB group, the discrepancy was 47%, with a substantial risk of underestimating the malignant potential of the tumor. Hoeber [29] reported superior accuracy in biopsying soft tissue sarcomas (69–99%); however, in his paper, there is no mention of different results by histology subset. We did not find a correlation between ML misgrading and main clinicopathological characteristics (size, location, depth). Therefore, diagnostic inconsistencies could be related to tumor heterogeneity and/or the presence of neoadjuvant treatments.

We determined that neoadjuvant therapy is associated with downgrading in the high-grade group of patients. ML is highly sensitive to RT treatment [5,8]. In fact, we demonstrated that patients treated with neoadjuvant RT had a higher probability of downgrading compared to patients treated with neoadjuvant ChT. Previous studies have looked into the accuracy of biopsy techniques in terms of determining malignancy, grade, and subtype [29,35,36], but none have looked at the effect of neoadjuvant treatment. A reliable assessment of the percentage of hypercellularity would likely require adequate and extended sampling of the tumor, as performed in an open biopsy [30], or the definition of new morphologic criteria related to the grade of malignancy.

The reported sensitivity for TCB compared to the final specimen biopsies is in the range of 82% and 92%, with a negative predictive value between 76% and 91% [29]. In the present study, the specificity of TCBs was 100%, a result consistent with those previously reported [18]. The sensitivity was 57%, indicating that there are as many true positives as there are false negatives (a = c) and that the test is not useful in detecting disease. The diagnostic accuracy of MLS biopsy is multifactorial, and the sensitivity of TCB in detecting high-grade lesions may depend either on technical or clinical features. Technical sensitivity correlates to the experience and ability of clinicians and to the tools available during the biopsy. The radiologist and the surgeon must be experts in sarcoma diagnosis to rigorously determine which portion of a tumor could better resemble its histology. Possibly, soft tissue tumors are an even easier sample to collect than other sarcomas, and TCB allows for collecting deeper parts of the mass in comparison to other techniques. We thus believe that the competence of the medical staff is determinant in increasing the sensitivity of high-grade lesion analysis. Since this study was performed in a highly specialized center for bone and soft tissue pathologies and considering that the medical staff was an expert in this field, we do not think that technical skills impacted the poor sensibility of the test. Clinical sensitivity, instead, is related to the material examined and to its quality. Underestimation of grade on TCB could be due to the lower quality of tissue sampled by TCB. The tissue obtained may not include the tumor’s growing edge, or there may be insufficient tumor present to complete a formal count. Myxoid histology is associated with decreased accuracy because of the presence of transitional areas between low- and high-grade histology. High-grade subtypes are even harder to diagnose since the use of different threshold values corresponding to hypercellular areas could influence the sensitivity of the test.

Some limitations must be acknowledged, including the study’s retrospective design; therefore, some data could be fragmentary and difficult to trace. Moreover, the small number of patients participating in the study and the large time of follow-up considered, which could be linked to the development of different therapeutic approaches, are other limitations. Despite the above-mentioned limitations, based on our results and on previous reports, we believe that accurate diagnosis with TCB is not as simple as it may seem in this subset of malignancies. Yet, future application of more advanced tools, such as the combination of imaging analysis (radiomics) and pathological (pathomics) features, the use of new clinicopathological scoring, or revision of the histology grading system will be decisive in improving prognosis in myxoid liposarcoma [12].

## 5. Conclusions

Regardless of the clinicopathological features, in a clinical setting with a multimodal approach, discrepancies in liposarcoma grading using TCB may occur in up to 36% of the cases with a determinant downgrading effect of preoperative chemotherapy and/or RT. Nevertheless, in cases of misdiagnosis of high-grade and low-grade tumors, overall survival is not affected because systemic treatment decision-making also includes other variables. In the absence of preoperative treatment, the sensitivity of TCB was 57%, and specificity was 100%.

## Figures and Tables

**Figure 1 cancers-15-01391-f001:**
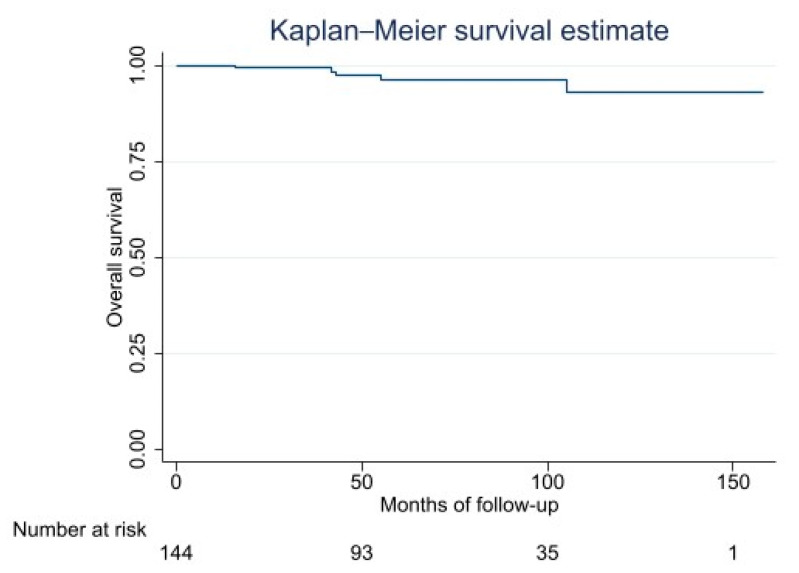
Kaplan–Meier Overall Survival Curve.

**Figure 2 cancers-15-01391-f002:**
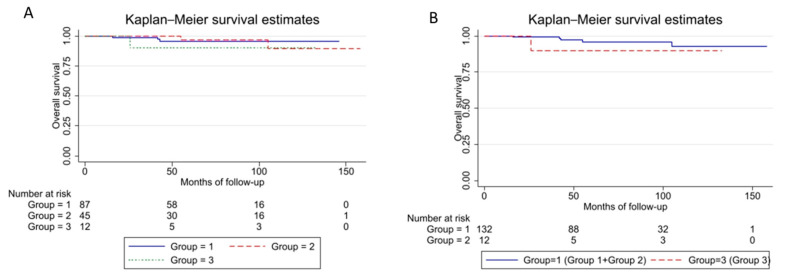
Kaplan–Meier curve stratified by histologic grade group. (**A**) Group 1 vs. Group 3 vs. Group 3, Log-rank test Pr > chi2 = 0.6061. (**B**) Group 1 + Group 2 vs. Group 3, Log-rank test pr = 0.3237. Group 1 = High-grade group (high-grade TCB); Group 2 = Low-grade group (low-grade TCB and final biopsy); Group 3 = Upgrade group (low-grade TCB and high-grade final biopsy). Log-rank test Pr > chi2 = 0.6061.

**Table 1 cancers-15-01391-t001:** Main Clinicopathological Features and Therapeutic Approaches.

Factor	Number of Patients	%
Patients	144	100
Gender		
Male	91	63
Female	53	37
Location		
Thigh	100	69
Lower limb	33	21
Buttock	10	7
Arm	1	1
Depth		
Deep	136	95
Superficial	8	5
Tumor Size		
>10 cm	83	58
5–10 cm	55	38
<5 cm	6	4
Preoperative Grade		
High	87	60
Low	57	40
Postoperative Grade		
High	59	41
Low	85	59
Surgery		
Excision	142	99
Amputation	2	1
Margin		
Wide/Radical (R0)	104	72
Marginal (R1)	35	25
Intralesional (R2)	5	3
Radiotherapy		
Preoperative	92	64
Postoperative	31	21
None	21	15
Chemotherapy		
Preoperative	46	32
Postoperative	20	14
Pre- and postop.	2	1
None	76	53

**Table 2 cancers-15-01391-t002:** Grade Diagnosis of The Core Needle Biopsy Specimen and Surgical Specimen.

		Surgical Specimen	
Core Needle Biopsy	Tot	High grade	Low grade
Group A (High Grade)	87	47 (54%)	40 (46%)
Group B (Low Grade)	57	12 (21%)	45 (79%)

**Table 3 cancers-15-01391-t003:** Factors Correlated to Histological Downgrading Limited to Group A.

					Univariate Analysis		
Factor		N	Discordant (N)	Discordant (%)	OR	*p*-Value	95% CI
Age	<50	47	23	57	0.84	0.687	0.35–1.9
	>50	40	17	43			
Location	Thigh (ref)	55	24	60			
	Lower limb	25	13	33	1.39	0.488	0.54–3.61
	Buttock Arm	6 1	2 1	5 2	0.64	0.630	0.10–3.8
Size	>10 cm (ref)	51	26	65			
	5 < to < 10 cm	31	11	28	0.53	0.174	0.21–1.32
	<5 cm	5	3	7	1.44	0.701	0.22–9.37
Depth	Deep	82	39	98	3.6	0.25	0.38–33.8
	Superficial	5	1	2			
Neoadjuvant therapy	ChT (ref)	12	3	7.5			
	RT	23	14	35	4.66	0.052	0.9–21
	ChT + RT	36	23	57.5	5.30	0.026	1.3–24
	None	16	0				

Reference (ref); 95% confidence interval (95% CI); radiotherapy (RT); chemotherapy (ChT).

**Table 4 cancers-15-01391-t004:** Overall Survival: Kaplan–Meier Survival Estimation.

	Survival Function	95% Confidence Interval
2 years	99	94–99
5 years	96	90–98
10 years	93	81–97

**Table 5 cancers-15-01391-t005:** OS Rates Stratified by Histologic Grade Group.

	Factor (N)	5-Years OS (%)	95% CI	10-Years OS (%)	95% CI
Grade group	1 (87)	96	87–98	96	87–98
	2 (45)	97	78–99	90	62–97
	3 (12)	90	47–98	90	47–98
Grade group	1 + 2 (132)	96	89–98	93	80–97
	3 (12)	90	47–99	90	47–99

Group 1 = High-grade group (high-grade TCB); Group 2 = Low-grade group (low-grade TCB and final biopsy); Group 3 = Upgrade group (low-grade TCB and high-grade final biopsy). OS: overall survival; 95% Interval of confidence (95% CI).

**Table 6 cancers-15-01391-t006:** Accuracy of Tru-Cut^®^ Biopsy Techniques Determining Malignancy When Compared to The Final Diagnosis. Analysis of 40 MLS Not Treated with Neoadjuvant Therapy.

	True Diagnosis (Surgical Specimen Histology)		
	High-Grade	Low-Grade	Tot
TCB			
High-grade	16 (TP)	0 (FP)	16
Low-grade	12 (FN)	12 (TN)	24
Tot	28	12	40

Tru-Cut^®^ biopsy (TCB), TP: true positive; FN: false negative; TN: true negative. Sensitivity: 57%, 95% CI (37–76). Specificity: 100%, 95% CI (74–100). Predictive values of positive results: 100%, 95% CI (79–100). Predictive values of negative results: 50%, 95% CI (29–71).

## Data Availability

The data presented in this study are available upon request from the corresponding author. The data are not publicly available due to privacy or ethical restrictions.

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
