# Peer review of "Sensitivity, Specificity, and Predictive Values of Tru-Cut® Biopsy in Grading Primary Localized Myxoid Liposarcomas of the Extremities"

_cancers, 2023, doi:10.3390/cancers15051391_

Round 1
Reviewer 1 Report
Would include a discussion about how grade dx on CNBx may change preoperative treatment decisions for MLPS, as this is the whole crux of their research question (high grade gets rad +/- chemo; is the timing of this decision preop before you have final pathology making a difference in patient outcomes?). This sort of study has been done before in other types of cancers (ie, receptor testing in breast cancer), so this can help explain why it is worth looking into in MLPS.
Would include exclusion criteria- why were only 144/150 patients included?
Lines 178-180- clarify wide margins, marginal margins and intralesional margins- does this correspond to R0 (microscopically negative), R1 (macroscopically negative, microscopically positive) and R2 (grossly positive)?
Line 242- change “excision biopsy” to “excisional biopsy”
How many samples were collected per patient and were they adequate for pathologic analysis?
Please add time units for Kaplan-meier curves
Would include limitations- mentions study size in passing but no discrete section
Line 293- not sure what is meant by “definition of new malignant criteria” as a means of improving accuracy in percentage of hypercellularity assessment
While it is alluded to in lines 268-273, a discrete discussion about why the sensitivity is so poor in detecting high grade lesions would be beneficial
Line 313- change “sensibility and specificity” to “sensitivity”
Reviewer 2 Report
This paper presents a comparison of histological grading of localised limb myxoid liposarcoma at initial trucut biopsy and at definitive surgery. The most interesting group are those that have no neoadjuvant treatment where the findings show that a proportion of high grade cases are missed. However in 12 patients who were upgraded from low grade to high grade after surgery, outcomes were not adversely affected.
I think the results of the study are unsurprising but nevertheless may be of use to clinicians. It not very clear what changes in practice would be recommended by the authors as a result of their findings and it would be helpful to discuss more explicitly.
I have a number of minor comments:
Line 73 - is reference 11 the correct reference for WHO classification?
Section 3: the term "leg" as a disease site needs to be reviewed. The leg includes both the thigh and lower leg (calf/shin). Can they be more specific
Line 193 - please rephrase the subtitle "determination of dignity" - dignity is not the right word here
Fig 1: the axes of the survival curve need labelling. Overall survival on y axis. Is analysis time in months? Same comment for Fig 2.
Table 5_ can you do a simple statistical analysis on the OS rates of grp 3 v Gp 1 and 2?
Table 6- typo in Diagnosis
Round 2
Reviewer 1 Report
comments / suggested edits nicely addressed